# FL Games: A federated learning framework for distribution shifts

## Abstract

Federated learning aims to train models for data that is distributed across clients, under the orchestration of a server. However, participating clients typically each hold data from a different distribution, which can yield catastrophic generalization on data from a different client, which represents a new domain. In this work, we argue that in order to generalize better across non-i.i.d. clients, it is imperative to only learn correlations that are stable and invariant across domains. We propose FL GAMES, a game-theoretic framework for federated learning that learns causal features that are invariant across clients. While training to achieve the Nash equilibrium, the traditional best response strategy suffers from high-frequency oscillations. We demonstrate that FL GAMES effectively resolves this challenge and exhibits smooth performance curves. Further, FL GAMES scales well in the number of clients, requires significantly fewer communication rounds, and is agnostic to device heterogeneity. Through empirical evaluation, we demonstrate that FL GAMES achieves high out-of-distribution performance on various benchmarks.

## 1 Introduction

With the rapid advance in technology and growing prevalence of smart devices, Federated Learning (FL) has emerged as an attractive distributed learning paradigm for machine learning models over networks of computers (Kairouz et al., 2019; Li et al., 2020; Bonawitz et al., 2019). In FL, multiple sites with local data, often known as *clients*, collaborate to jointly train a shared model under the orchestration of a central hub called the *server* while keeping their data private.

While FL serves as an attractive alternative to centralized training because the client data does not need to move to the server, there are several challenges associated with its optimization: 1) *statistical heterogeneity* across clients; 2) *massively distributed with limited communication* i.e., a large number of client devices with only a small subset of active clients at any given time (McMahan et al., 2017; Li et al., 2020). One of the most popular algorithms in this setup, Federated Averaging (FEDAVG) (McMahan et al., 2017), allows multiple updates at each site prior to communicating updates with the server. While this technique delivers huge communication gains in i.i.d. (independent and identically distributed) settings, its performance on non-i.i.d. clients is an active area of research. As shown by Karimireddy et al. (2020), client heterogeneity has direct implications on the convergence of FEDAVG since it introduces a *drift* in the updates of each client with respect to the server model. Yao et al. (2019) describes the same phenomena by arguing that multiple steps of updates at each client cause *gradient bias* towards model aggregation. While recent works including Li et al. (2019); Karimireddy et al. (2020); Yu et al. (2019); Wang et al. (2020); Li et al. (2020); Lin et al. (2020); Li & Wang (2019); Zhu et al. (2021) have tried to address client heterogeneity through constrained gradient optimization and knowledge distillation, most did not tackle the underlying distribution shift. These methods mostly adapt variance reduction techniques such as Stochastic Variance Reduction Gradients (SVRG) (Johnson & Zhang, 2013) to FL. The bias among clients is reduced by constraining the updates of each client with respect to the aggregated gradients of all other clients. These methods can at

best generalize to interpolated domains and fail to extrapolate well, i.e., generalize to newly extrapolated domains [1].

According to Pearl (2018) and Schölkopf (2019), in order to build robust systems that generalize well outside of their training environment, learning algorithms should be equipped with causal reasoning tools. Over the past year, there has been a surge in interest in bringing the machinery of causality into machine learning (Arjovsky et al., 2019; Ahuja et al., 2020; Schölkopf, 2019; Ahuja et al., 2021b; Parascandolo et al., 2020; Robey et al., 2021; Krueger et al., 2021; Rahimian & Mehrotra, 2019). All these approaches have focused on learning causal dependencies, which are stable across training environments and further estimate a truly invariant and causal predictor. However, despite their success, they suffer from several key limitations rendering them unsuitable for deployment in a real-world setup.

Since FL typically consists of a large number of clients, it is natural for data at each client to represent different annotation tools, measuring circumstances, experimental environments, and external interventions. Predictive models trained on such datasets could simply rely on spurious correlations to improve their in-distribution, i.i.d. performance. Inspired by this idea and by the recent progress in causal machine learning, we draw connections between OOD generalization and robustness across heterogeneous clients in FL. In particular, we consider IRM Games (Ahuja et al., 2020) from the OOD generalization literature since its formulation shows resemblance to the standard FL setup. However, as discussed above, IRM Games too encounters a few fundamental challenges not just specific to FL but also in a generic ML framework.

- **Sequential dependency.** The underlying game theoretic algorithm in IRM Games is inherently sequential i.e. at any instance, exactly one player chooses their optimal action. This causes the time complexity of the algorithm to scale linearly with the number of environments.

- **Oscillations.** IRM Games exhibits large oscillations in the performance metrics as the training progresses. The high frequency of these oscillations makes it difficult to define a valid stopping criterion.

- **Convergence speed.** The convergence rate of IRM Games is slow and hence directly impacts systems with speed or communication cost as a primary bottleneck.

In this study, we take a step towards fixing these limitations and addressing the challenge of client heterogeneity under distribution shifts in FL from a causal viewpoint. Specifically, drawing inspiration from IRM Games, we propose Federated Learning Games (FL Games) for learning causal representations which are stable across clients and further enhancing the generalizability of the trained model across unseen testing domains. Apart from effectively learning causal features across clients, our algorithm FL Games also addresses each of the above challenges, hence providing a robust, efficient, and scalable solution. We summarize our main contributions below.

- We propose a new framework called FL Games for learning causal representations that are invariant across clients in a federated learning setup.

- Inspired by the game theory literature, we equip our algorithm to allow parallel updates across clients, further resulting in superior scalability.

- Using ensembles over the client's historical actions, we demonstrate that FL Games appreciably smoothens the observed oscillations.

- By increasing the local computation at each client, we show that FL Games exhibits high communication efficiency.

- Empirically, we show that the performance of the invariant predictors found by our approach on unseen OOD clients improves significantly over state-of-art prior work.

---

[1]Similar to Krueger et al. (2021), we define interpolated domains as the domains which fall within the convex hull of training domains and extrapolated domains as those that fall outside of that convex hull.

## 2 Related work

**Federated learning (FL)**. A major challenge in federated learning is data heterogeneity across clients where the local optima at each client may be far from that of the global optima in the parameter space. This causes a *drift* in the local updates of each client with respect to the server aggregated parameters and further results in slow and unstable convergence (Karimireddy et al., 2020). Recent works have shown FEDAVG to be vulnerable in such heterogeneous settings (Zhao et al., 2018). A subset of these works that explicitly constrains gradients for bias removal are called extra gradient methods. Among these methods, FEDPROX (Li et al., 2020) imposes a quadratic penalty over the distance between server and client parameters which impedes model plasticity. Others use a form of variance reduction techniques such as SVRG (Johnson & Zhang, 2013) to regularize the client updates with respect to the gradients of other clients (Acar et al., 2021; Li et al., 2019; Karimireddy et al., 2020; Liang et al., 2019; Zhang et al., 2020; Konečný et al., 2016). Karimireddy et al. (2020) communicates to the server an additional set of variables known as control variates which contain the estimate of the update direction for both the server and the clients. Using these control variates, the drift at each client is estimated and used to correct the local updates. On the other hand, Acar et al. (2021) estimates the drift for each client on the server and corrects the server updates. By doing so, they avoid using control variates and consume less communication bandwidth. The general strategy for variance reduction methods is to estimate client drift using gradients of other clients, and then constrain the learning objective to reduce the drift. The extra gradient methods are not explicitly optimized to discover causal features and thus may fail with out-of-distribution examples outside the aggregated distribution of clients.

To date, only two scientific works, Francis et al. (2021); Tenison et al. (2021) have incorporated the learning of invariant predictors in order to achieve strong generalization in FL. The former adapts masked gradients as in Parascandolo et al. (2020) and the latter builds on IRM to exploit invariance and improve leakage protection in FL. While IRM lacks theoretical convergence guarantees, failure modes of Parascandolo et al. (2020) like the formation of dead zones and high sensitivity to small perturbations (Shahtalebi et al., 2021) are also issues when it is applied in FL, rendering it unreliable.

**Out-of-distribution (OOD) generalization.** Generalization under distributional shift is one of the major challenges faced by machine learning systems, limiting their application in the real world. Recent research including Arjovsky et al. (2019); Ahuja et al. (2020); Schölkopf (2019); Ahuja et al. (2021b); Parascandolo et al. (2020); Robey et al. (2021); Krueger et al. (2021); Rahimian & Mehrotra (2019); Xie et al. (2020); Yao et al. (2022); Ahuja et al. (2021a) have tried to address this challenge by proposing alternative objectives for training mechanisms that are invariant across training environments. IRM (Arjovsky et al., 2019) proposes finding a representation $\phi(X)$ that has good prediction abilities and also elicits an invariant predictor across environments. Works like Krueger et al. (2021); Xie et al. (2020) propose a penalty that is a function of the variance of training risks. Ahuja et al. (2020) reformulates IRM as finding the Nash equilibrium of an ensemble game played among environments. Mahajan et al. (2021) argues that learning invariant representations for inputs derived from the same object. Recently, Robey et al. (2021) proposed Model-Based Domain Generalization, which enforces invariance to the underlying transformations of data. Another line of work (Rosenfeld et al., 2020; Kamath et al., 2021; Ahuja et al., 2021a) has theoretically analyzed failures of IRM.

## 3 Background

### 3.1 Federated Learning

Federated learning methods involve a cloud server coordinating among multiple client devices to jointly train a global model without sharing data across clients. Denote $\mathcal{S}$ as the set of client devices where $|\mathcal{S}| = m$. Let $N_k$ denote the number of data samples at client device $k$, and $\mathcal{D}_k = \{(x_i^k, y_i^k)\}_{i=1}^{N_k}$ as the corresponding labelled dataset. Mathematically, the objective of an FL is to approximately minimize the global loss

$$F(w) := \sum_{k=1}^{m} \frac{1}{m} \sum_{i=1}^{N_k} \frac{1}{N_k} \ell(x_i^k, y_i^k, w), \tag{1}$$

where $\ell$ is the loss function and $w$ is the model parameter.

One of the most popular methods in FL is Federated Averaging (FEDAVG), where each client performs $E$ local updates before communicating its weights with the server. FEDAVG becomes equivalent to FEDSGD for $E = 1$ wherein weights are communicated after every local update. For each client device $k$, FEDAVGinitializes its corresponding device model $w_k^0$. Consequently, in round $t$, each device undergoes a local update on its dataset according to the following $w_k^{t+1} \leftarrow w_k^t - \eta^k \nabla \ell(B_i^k, w_k^t), \forall B_i^k \subseteq \mathcal{D}_k$ where $B_i^k$ is a sampled mini-batch from $D_k$ at the $i$th step. All clients' model parameters $\{w_k^{t+1}\}_{k \in \mathcal{S}}$ are then sent to the cloud server which performs a weighted average to update the global model $w^{t+1}$ as

$$w^{t+1} \leftarrow \frac{1}{|\mathcal{S}|} \sum_{k \in \mathcal{S}} \frac{N_k}{N} w_k^t,$$

where $N_k$ is the number of samples at client device $k$ and $N$ is the total number of samples from all clients ($N = \sum_{k=1}^{m} N_k$). This aggregated global model is broadcasted with all clients and the above process is repeated till convergence.

### 3.2 Invariant Risk Minimization Games

Consider a setup comprising datasets $\mathcal{D}_k = \{(x_i^k, y_i^k)\}_{i=1}^{N_k}$ from multiple training environments, $k \in \mathcal{E}_{tr}$ with $N_k$ being the number of samples at environment $k$ and $\mathcal{E}_{tr}$ as the index set of training environments. The aim of Invariant Risk Minimization (IRM) (Arjovsky et al., 2019) is to jointly train across all these environments and learn a robust set of parameters $\theta$ that generalize well to unseen (test) environments $\mathcal{E}_{all} \supset \mathcal{E}_{tr}$. The risk of a predictor $f$ at each environment can be mathematically represented as $R^k(w \circ \phi) = E_{(x,y) \sim \mathcal{D}_k} f_\theta(x, y)$ where $f_\theta = w \circ \phi$ is the composition of a feature extraction function $\phi : \mathcal{X} \to \mathcal{Z} \subseteq \mathbb{R}^d$ and a predictor network, $w : \mathcal{Z} \to \mathbb{R}^k$ where $\mathcal{X}$ denotes the input space, $\mathcal{Z}$ denotes the representation space and, $k$ is the number of classes.

**Empirical Risk Minimization (ERM)** aims to minimize the average of the losses across all environments. Mathematically, the ERM objective can be formulated as $R^{\text{ERM}}(\theta) = E_{(x,y) \sim \cup_{k \in \mathcal{E}_{tr}} \mathcal{D}_k} f_\theta^k(x, y)$, where $f_\theta^k$ is the composition of feature extractor $\phi$ and the predictor network for environment $k$ i.e. $w^k$. As shown in Arjovsky et al. (2019), ERM fails to generalize to novel domains, which have significant distribution shifts as compared to the training environments.

**Invariant Risk Minimization (IRM)** instead aims to capture invariant representations $\phi$ such that the optimal predictor $w$ given $\phi$ is the same across all training environments. Mathematically, they formulate the objective as a bi-level optimization problem

$$\min_{\phi \in \mathcal{H}_\phi, w \in \mathcal{H}_w} \sum_{k \in \mathcal{E}_{tr}} R^k(w \circ \phi) \text{ s.t. } w \in \arg\min_{w \in \mathcal{H}_w} R^k(w \circ \phi), \forall k \in \mathcal{E}_{tr} \tag{2}$$

where $\mathcal{H}_\phi$, $\mathcal{H}_w$ are the hypothesis sets for feature extractors and predictors, respectively. Since each constraint calls an inner optimization routine, IRM approximates this challenging optimization problem by fixing the predictor $w$ to a scalar.

**Invariant Risk Minimization Games (IRM Games)** is an algorithm based on an alternate game theoretic reformulation of the optimization objective in equation 2. It endows each environment with its own predictor $w^k \in \mathcal{H}_w$ and aims to train an ensemble model $w^{av}(z) = \frac{1}{|\mathcal{E}_{tr}|} \sum_{k=1}^{|\mathcal{E}_{tr}|} w^k(z)$ for each $z \in \mathcal{Z}$ s.t. $w^{av}$ satisfies the following optimization problem

$$\min_{w^{av}, \phi \in \mathcal{H}_\phi} \sum_k R^k(w^{av} \circ \phi)$$

$$\text{s.t. } w^k \in \underset{w'_k \in \mathcal{H}_w}{\arg\min} R^k\left(\frac{1}{|\mathcal{E}_{tr}|}(w'_k + \sum_{q \in \mathcal{E}_{tr}, q \neq k} w^q) \circ \phi\right), \forall k \in \mathcal{E}_{tr} \tag{3}$$

The constraint in equation 3 is equivalent to the Nash equilibrium of a game with each environment $k$ as a player with action $w^k$, playing to maximize its utility $R^k(w^{av}, \phi)$. While there are different algorithms in the game-theoretic literature to compute the Nash equilibrium, the resultant non-zero sum continuous game is solved using the best response dynamics (BRD) with clockwise updates and is referred to as V-IRM GAMES. In this training paradigm, players take turns according to a fixed cyclic order, and only one player is allowed to change its action at any given time (for more details, refer to the supplement). Fixing $\phi$ to an identity map in V-IRM GAMES is also shown to be very effective and is called F-IRM GAMES.

## 4 Federated Learning Games (FL Games)

OOD generalization is often typified using the notion of data-generating environments. Arjovsky et al. (2019) formalizes an environment as a data-generating distribution representing a particular location, time, context, circumstances, and so forth. Distinct environments are assumed to share some overlapping causal features such that the corresponding causal mechanisms are invariant across environments, but the distribution of some of the causal variables may vary. Spurious variables denote the unstable features which vary across environments. This concept of data-generating environments can be related to FL by considering each client as producing data generated from a different environment. However, despite this equivalence, existing OOD generalization techniques can not be directly applied to FL. Apart from the FL-specific challenges, these approaches also suffer from several key limitations in non-FL domains (Rosenfeld et al., 2020; Nagarajan et al., 2020), further rendering them unfit for practical deployment. As a consequence, developing causal inference models for FL that are inspired by invariant prediction in OOD generalization, are bound to inherit the failures of the latter.

In this work, we consider one such popular OOD generalization technique, IRM GAMES, as its formulation bears a close resemblance to a standard FL setup. However, as discussed, IRM GAMES, too, suffers from various challenges, which impede its deployment in a generic ML framework, specifically in FL. In the following section, we elaborate on each of these limitations and discuss the corresponding modifications required to overcome them. Further, inspired by the game theoretic formulation of IRM GAMES, we propose FL GAMES which forfeits its failures and can recover the causal mechanisms of the targets, while also providing robustness to changes in the distribution across clients.

### 4.1 Challenges in Federated Learning

**Data Privacy.** Consider a FL system with $m$ client devices, $\mathcal{S} = \{1, 2, ..., m\}$. Let $N_k$ denote the number of data samples at client device $k$, and $\mathcal{D}_k = \{(x_i^k, y_i^k)\}_{i=1}^{N_k}$ as the corresponding labelled dataset. The constraint of each environment in IRM GAMES can be used to formulate the local objective of each client. In particular, each client $k \in \mathcal{S}$ now serves as a player, competing to learn $w^k \in \mathcal{H}_w$ by optimizing its local objective, i.e.

$$w^k \in \underset{w'_k \in \mathcal{H}_w}{\arg\min} R^k\left(\frac{1}{|\mathcal{S}|}(w'_k + \sum_{\substack{q \in \mathcal{S} \\ q \neq k}} w^q) \circ \phi\right), \forall w'_k \in \mathcal{H}_w \tag{4}$$

However, the upper-level objective of IRM GAMES requires optimization over the dataset pooled together from all environments. Centrally hosting the data on the server or sharing it across clients contradicts the objective of FL. Hence, we propose using FEDSGD (McMahan et al., 2017) to optimize $\phi$. Specifically, each client $k$ computes and broadcasts gradients with respect to $\phi$

$$g^k = \nabla R^k\left(\frac{1}{|\mathcal{S}|}(w'_k + \sum_{q \neq k} w^k) \circ \phi\right)$$

to the server, which then aggregates these gradients and applies the update rule $\phi_{t+1} = \phi_t - \eta \sum_{k \in \mathcal{S}} \frac{N_k}{N} g^k$ where $g^k$ is computed over $C\%$ batches locally and $N = \sum_{k \in S} N_k$.

Similar to IRM GAMES, we call this approach V-FL GAMES. The variant with $\phi = I$ is called F-FL GAMES.

**Sequential dependency** As discussed, IRM GAMES poses IRM objective as finding the Nash equilibrium of an ensemble game across environments and adopting the classic best response dynamics (BRD) algorithm to compute it. This approach is based on playing clockwise sequences wherein players take turns in a fixed cyclic order, with only one player being allowed to change their action at any given time $t$ (Details in the supplement). In order to choose its optimal action for the first time, the last scheduled player $N$ has to wait for all the remaining players from $1, 2, ...N-1$ to play their strategies. This linear scaling of time complexity with the number of players poses a major challenge in solving the game in FL.

By definition, in classic BRD, the best responses of any player determine the best responses of the remaining players. Thus, in a distributed learning paradigm, the best responses of each client (player) need to be transmitted to all the other clients. This is infeasible from a practical standpoint as clients are usually based on slow or expensive connections, and message transmissions can frequently get delayed or result in information loss. As shown on lines 20 and 26 (green) of Algorithm 1, we modify the classic BRD algorithm by allowing simultaneous updates at any given round $t$. However, a client now best responds to the optimal actions played by its opponent clients in round $t-1$ instead of $t$. We refer to this approximation of BRD in F-FL GAMES and V-FL GAMES as *parallelized* F-FL GAMES and, *parallelized* V-FL GAMES respectively.

**Oscillations.** As demonstrated in Ahuja et al. (2020), when a neural network is trained using the IRM GAMES objective (equation 3), the training accuracy initially stabilizes at a high value and eventually starts to oscillate. The setup, over which these observations are made, involves two training environments with varying degrees of spurious correlation. The environments are constructed so that the degree of correlation of color with the target label is very high. The explanation for these oscillations attributes to the significant difference among the data of the training environments. In particular, after a few steps of training, the individual model of the environment with higher spurious correlation (say $\mathcal{E}_1$) is positively correlated with the color while the other is negatively correlated. When it is the turn of the former environment to play its optimal strategy, it tries to exploit the spurious correlation in its data and increase the weights of the neurons which are indicative of color. On the contrary, the latter tries to decrease the weights of features associated with color since the errors that backpropagate are computed over the data for which exploiting spurious correlation does not work (say $\mathcal{E}_2$). This continuous swing and sway among individual models result in oscillations.

Despite the promising results, with the model's performance metrics oscillating to and from at each step, defining a reasonable stopping criterion becomes challenging. As shown in various game theoretic literature including Herings & Predtetchinski (2017); Barron et al. (2010); Fudenberg et al. (1998); Ge et al. (2018), BRD can often oscillate. Computing the Nash equilibrium for general games is non-trivial and is only possible for a certain class of games (e.g., concave games) (Zhou et al., 2017). Thus, rather than alleviating oscillations completely, we propose solutions to reduce them significantly to better target valid stopping points. We propose a two-way ensemble approach wherein apart from maintaining an ensemble across clients ($w^{av}$), each client $k$ also responds to the ensemble of historical models (memory) of its opponents. Intuitively, a moving ensemble over the historical models acts as a smoothing filter, which helps to reduce drastic variations in strategies.

Based on the above motivation, we reformulate the optimization objective of each client (Equation 4) to adapt the two-way ensemble learning mechanism (refer to line 4 (red) in Algorithm 1). Formally, we maintain queues (a.k.a. buffer) at each client, which stores its historically played actions. In each iteration, a client best responds to a uniform distribution over the past strategies of its opponents. The global objective at the server remains unchanged. Mathematically, the new local objective of each client $k \in \mathcal{E}_{tr}$ can be stated as

$$w^k \in \underset{w_k' \in \mathcal{H}_w}{\arg\min} R^k \left( \frac{1}{|\mathcal{S}|} (w_k' + \sum_{\substack{q \in \mathcal{S} \\ q \neq k}} w^q + \sum_{\substack{p \in \mathcal{S} \\ p \neq k}} \frac{1}{|\mathcal{B}_p|} \sum_{j=1}^{|\mathcal{B}_p|} w_j^p) \circ \phi \right) \tag{5}$$

where $\mathcal{B}_q$ denotes the buffer at client $q$ and $w_j^q$ denotes the $j$th historical model of client $q$. We use the same buffer size for all clients. Moreover, as the buffer reaches its capacity, it is renewed based on a first in first out (FIFO) manner. Note that this approach does not result in any communication overhead since a running sum over historical strategies can be calculated in $\mathcal{O}(1)$ time by maintaining a prefix sum. This variant is called F-FL GAMES (SMOOTH) or V-FL GAMES (SMOOTH) based on the constraint on $\phi$.

---

**Algorithm 1** *Parallelized* FL GAMES (SMOOTH+FAST)

---

1: **Notations:**  $\mathcal{S}$ is the set of $N$ clients; $\mathcal{B}_k$ and $\mathcal{P}_k$ denote the buffer and information set containing copies of $\mathcal{B}_i, \forall i \neq k \in \mathcal{S}$ at client, $k$ respectively.

2: **PredictorUpdate(k):**

3:      /* Two-way ensemble game to update predictor at each client $k$ */

4:      $w_k \leftarrow \text{SGD} \left[ \ell_k \left\{ \frac{1}{|\mathcal{S}|} (w_k' + \sum_{\substack{q \in \mathcal{S} \\ q \neq k}} w^q + \boxed{\sum_{\substack{p \in \mathcal{S} \\ p \neq k}} \frac{1}{|\mathcal{B}_p|} \sum_{j=1}^{|\mathcal{B}_p|} w_j^p}) \circ \phi \right\} \right]$

5:      Insert $w_k$ to $\mathcal{B}_k$, discard oldest model in $\mathcal{B}_k$ if full

6:      return $w_k$

7: **RepresentationUpdate(k):**

8:      /* Gradient Descent (GD) over entire local dataset at client $k$ */

9:      **for** every batch $b \in \mathcal{B}$ **do**

10:          Compute $\nabla \ell_k(w_{\text{cur}}^{\text{av}} \circ \phi_{\text{cur}}; b)$; Add in $\nabla \phi_k$

11:      return $\nabla \phi_k$

12: **Server executes:**

13:      Initialize $w_k, \forall k \in \mathcal{S}, w_{\text{curr}}^{\text{av}}$ and $\phi_{\text{cur}}$

14:      **while** round $\leq$ max-round **do**

15:          /* Update representation $\phi$ at even round parity */

16:          **if** round is even **then**

17:              **if** Fixed-Phi **then**

18:                  $\phi_{\text{cur}} = I$

19:              **if** Variable-Phi **then**

20:                  **for** each client $k \in \mathcal{S}$ in parallel **do**

21:                      $\nabla \phi_k = \text{RepresentationUpdate(k)}$

22:                  /* Update representation $\phi$ */

23:                  $\phi_{\text{next}} = \phi_{\text{cur}} - \eta \left( \sum_{k \in \mathcal{S}} \frac{N_k}{\sum_{j \in \mathcal{S}} N_j} \nabla \phi_k \right)$

24:                  $\phi_{\text{cur}} = \phi_{\text{next}}$

25:          **else**

26:              **for** each client $k \in \mathcal{S}$ in parallel **do**

27:                  $w_{\text{curr}}^k \leftarrow \text{PredictorUpdate(k)}$

28:              /* Client $k$ updates its information set $\mathcal{P}_k$ by updating copies of predictors of other clients */

29:              Communicate $\forall k, \mathcal{P}_k \leftarrow \{w_i, \forall i \neq k \in S\}$

30:          round $\leftarrow$ round $+ 1$

31:          $w_{\text{curr}}^{\text{av}} = \frac{1}{N} \sum_{k \in \mathcal{S}} w_{\text{curr}}^k$

---

**Convergence speed.** As discussed, FL GAMES has two variants: F-FL GAMES and, V-FL GAMES, with the former being an approximation of the latter ($\phi = I$, the identity matrix). While both approaches exhibit superior performance on a variety of benchmarks, the latter has shown its success in a variety of large-scale tasks like language modeling (Peyrard et al., 2021). Despite being theoretically grounded, V-IRM GAMES suffers from slower convergence due to an additional round for optimization of $\phi$.

Typically, clients (e.g., mobile devices) possess fast processors and computational resources and have datasets that are much smaller compared to the total dataset size. Hence, utilizing additional local computation is essentially free compared to communicating with the server. To improve the efficiency of our algorithm, we propose replacing the stochastic gradient descent (SGD) over $\phi$ by full-batch gradient descent (GD) (line 9 (yellow) of Algorithm 1). This allows $\phi$ to be updated according to gradients accumulated across the entire dataset, as opposed to gradient step over a mini-batch. Intuitively, now at each gradient step, the resultant $\phi$ takes large steps in the direction of its global optimum, resulting in fast and stable training. Note that the classifiers at each client are still updated over one mini-batch. This variant of FL GAMES is referred to V-FL GAMES (FAST).

## 5 Experiments and Results

### 5.1 Datasets

In Ahuja et al. (2020), IRM GAMES was tested over a variety of benchmarks which were synthetically constructed to incorporate color as a spurious feature. These included COLORED MNIST, COLORED FASHION MNIST, and COLORED DSPRITES dataset. We utilize the same datasets for our experiments. Additionally, we create another benchmark, SPURIOUS CIFAR10, with a data generating process resembling that of COLORED MNIST. In this dataset, instead of coloring the images to establish spurious correlation, we add small black patches at various locations in the image. These locations are spuriously correlated with the label. Details on each of the datasets can be found in the supplement. In all the results, we report the mean performance of various baselines over 5 runs. The performance of Oracle on each of these datasets is 75% for train and test sets.

**Terminologies:** In the following analysis, the terms 'Sequential' and 'Parallel' denote BRD with clockwise playing sequences and simultaneous updates, respectively (Lines 20 and 26 of Algorithm 1). We use FL GAMES as an umbrella term that constitutes all the discussed algorithmic modifications. F-FL GAMES and V-FL GAMES refer to the variants of FL GAMES, which abides by the data privacy constraints in FL. The approach used to smoothen out the oscillations (Line 4 of Algorithm 1) is denoted by F-FL GAMES (SMOOTH) or V-FL GAMES (SMOOTH) depending on the constraint on $\phi$. The fast variant with high convergence speed is typified as V-FL GAMES (SMOOTH+FAST) (Line 9 of Algorithm 1).

We compare these algorithmic variants across fixed and variable $\phi$ separately as shown in the Table **??**. Clearly, across all benchmarks, the FL baselines FEDSGD, FEDAVG (McMahan et al., 2017), FEDBN (Li et al., 2021) and FEDPROX (Li et al., 2020) are unable to generalize to the test set, with FEDBN exhibiting superior performance compared to the others. Intuitively, these approaches latch onto the spurious features to make predictions, hence leading to poor generalization over novel clients. Below, we study the performance of FL GAMES and its variants individually across the four benchmark datasets.

**Colored MNIST (Table 1).** We observe that our baseline approach i.e. sequential F-FL GAMES and V-FL GAMES achieve $66.56 \pm 1.58$ and $63.78 \pm 1.58$ percent testing accuracy, respectively. F-FL GAMES (SMOOTH) (Parallel) achieves the highest testing accuracy i.e $67.21 \pm 2.98$ amongst the fixed ($\phi = I$) variants of FL GAMES. Similarly, V-FL GAMES (Parallel) achieves the highest testing accuracy across the variable variants of FL GAMES i.e. $68.34 \pm 5.24$ percent. Clearly, all modifications in FL GAMES individually achieve high testing accuracy, hence eliminating the spurious correlations unlike state-of-the-art FL techniques.

**Colored Fashion MNIST (Table 1).** Similar to the results on COLORED MNIST, all modifications in FL GAMES achieve high testing accuracy. F-FL GAMES (SMOOTH) and V-FL GAMES achieve the highest test accuracy of $71.81 \pm 1.60$ and $69.90 \pm 1.31$ percent across the fixed and the variable variants, respectively. While the former was solely designed to smoothen the performance curves, it gives an additional benefit of higher OOD performance.

**Spurious CIFAR10 (Table 2).** F-FL GAMES (SMOOTH) (Parallel) and V-FL GAMES (SMOOTH+FAST) achieve the highest testing accuracy of $54.71 \pm 2.13$ and $50.94 \pm 3.28$ percent in the fixed and the variable categories respectively. Similar to the results on other datasets, all variants within FL GAMES achieve high testing accuracy.

Table 1: Comparison of methods in terms of training and testing accuracy (mean ± std deviation) across Colored MNIST and Colored Fashion MNIST. 'Seq.' and 'Par.' are abbreviations for sequential and parallel, respectively.

| | | | Colored MNIST | | Colored Fashion MNIST | |
|---|---|---|---|---|---|---|
| | | Algorithm | Train Accuracy | Test Accuracy | Train Accuracy | Test Accuracy |
| Baselines | | FedSGD | 84.88 ± 0.16 | 10.45 ± 0.60 | 83.49 ± 1.22 | 20.13 ± 8.06 |
| | | FedAVG | 84.45 ± 2.69 | 12.52 ± 4.34 | 86.23 ± 0.63 | 13.33 ± 2.07 |
| | | FedBN | 99.75 ± 0.11 | 47.16 ± 3.76 | 99.79 ± 0.17 | 41.24 ± 1.87 |
| | | FedPROX | 99.56 ± 0.38 | 29.31 ± 0.89 | 99.87 ± 0.12 | 29.41 ± 0.36 |
| Fixed | Seq. | F-FL Games | 55.76 ± 2.03 | 66.56 ± 1.58 | 75.13 ± 1.38 | 68.40 ± 1.83 |
| | | F-FL Games (Smooth) | 62.83 ± 5.06 | 66.83 ± 1.83 | 75.18 ± 0.37 | **71.81 ± 1.60** |
| | Par. | F-FL Games | 58.03 ± 6.22 | 67.14 ± 2.95 | 71.71 ± 8.23 | 69.73 ± 2.12 |
| | | F-FL Games (Smooth) | 61.07 ± 1.71 | **67.21 ± 2.98** | 72.81 ± 4.51 | 71.36 ± 4.19 |
| Variable | Seq. | V-FL Games | 56.40 ± 0.03 | 63.78 ± 1.58 | 69.90 ± 4.56 | **69.90 ± 1.31** |
| | | V-FL Games (Smooth+Fast) | 61.03 ± 3.11 | 65.81 ± 3.28 | 75.10 ± 0.48 | 69.85 ± 1.22 |
| | Par. | V-FL Games | 52.89 ± 8.03 | **68.34 ± 5.24** | 66.33 ± 9.39 | 69.85 ± 3.42 |
| | | V-FL Games (Smooth+Fast) | 63.11 ± 3.02 | 65.73 ± 1.53 | 71.89 ± 5.58 | 69.41 ± 5.49 |
| | | Optimal | 75 | 75 | 75 | 75 |

Table 2: Comparison of methods in terms of training and testing accuracy (mean ± std deviation) across Spurious CIFAR10 and Colored Dsprites. 'Seq.' and 'Par.' are abbreviations for sequential and parallel respectively.

| | | | Spurious CIFAR10 | | Colored Dsprites | |
|---|---|---|---|---|---|---|
| | | Algorithm | Train Accuracy | Test Accuracy | Train Accuracy | Test Accuracy |
| Baselines | | FedSGD | 84.79 ± 0.17 | 12.57 ± 0.55 | 99.15 ± 1.10 | 24.12 ± 2.00 |
| | | FedAVG | 85.41 ± 1.45 | 13.11 ± 1.82 | 99.21 ± 1.35 | 22.56 ± 2.34 |
| | | FedBN | 95.24 ± 2.34 | 25.16 ± 4.06 | 98.09 ± 1.45 | 25.19 ± 1.78 |
| | | FedPROX | 99.67 ± 0.13 | 24.76 ± 2.67 | 85.17 ± 1.95 | 11.12 ± 0.99 |
| Fixed | Seq. | F-FL Games | 50.36 ± 2.78 | 45.36 ± 4.33 | 53.98 ± 3.67 | 52.89 ± 4.41 |
| | | F-FL Games (Smooth) | 64.02 ± 2.08 | 45.54 ± 1.04 | 52.87 ± 3.30 | 61.45 ± 7.11 |
| | Par. | F-FL Games | 55.06 ± 2.04 | 52.07 ± 1.60 | 52.88 ± 2.78 | 56.50 ± 6.23 |
| | | F-FL Games (Smooth) | 56.98 ± 4.09 | **54.71 ± 2.13** | 53.65 ± 2.11 | **62.76 ± 5.97** |
| Variable | Seq. | V-FL Games | 61.72 ± 7.39 | 46.07 ± 6.01 | 51.36 ± 5.32 | 62.84 ± 7.20 |
| | | V-FL Games (Smooth+Fast) | 50.37 ± 4.97 | **50.94 ± 3.28** | 51.55 ± 3.20 | 68.23 ± 4.56 |
| | Par. | V-FL Games | 50.41 ± 3.31 | 50.43 ± 3.04 | 53.56 ± 4.91 | 65.87 ± 6.84 |
| | | V-FL Games (Smooth+Fast) | 45.83 ± 2.44 | 49.89 ± 5.66 | 54.25 ± 2.05 | **68.91 ± 6.47** |
| | | Optimal | 75 | 75 | 75 | 75 |

**Colored Dsprites (Table 2).** F-FL Games (Smooth) (Parallel) and V-FL Games (Smooth+Fast) achieve the highest testing accuracy of $62.76 \pm 5.97$ and $68.91 \pm 6.47$ percent across the fixed and variable variants of FL Games. Further, we discover that the mean performance of V-FL Games is superior to all algorithms with fixed representation ($\phi = I$). This accentuates the importance of V-FL Games over F-FL Games, especially over complex and larger datasets where learning $\phi$ becomes imperative.

In all the above experiments, both of our end approaches: *parallelized V-FL Games (Smooth+Fast)* and *parallelized* F-FL Games (Smooth) are able to perform better than or at par with the other variants. *These algorithms were primarily designed to overcome the challenges faced by causal FL systems while retaining their original ability to learn causal features.* Hence, the benefits provided by these approaches in terms of 1) robust predictions; 2) scalability; 3) fewer oscillations and 4) fast convergence are not at cost of performance. While the former is demonstrated by Tables 1 and 2, the latter three are detailed in Section 5.3.

[**Reviewers' #Y8Gz, #fzQW and #Bwep**]: A key observation across variants of FL Games is lower training accuracy compared to the testing accuracy. This phenomenon may be ascribed to the datasets utilized in the analysis, rather than an artifact of the algorithm. In particular, the construction of these datasets is such that the spurious feature captures information about the label, thereby providing additional insight beyond the predictive capability of the causal features. Consequently, any methodology that attempts to eliminate the spurious correlation may exhibit a negative correlation with this feature. Even a slight

negative reliance on such features may result in diminution of training accuracies. The attainment of high training accuracy under these circumstances is a challenging task and is contingent upon the nature of the datasets.

## 5.2 Interpretation of Learned Features

In order to explain the predictions of our parametric model, we use LIME (Ribeiro et al., 2016) to learn an interpretable model locally around each prediction. Specifically, using LIME, we construct an interpretable map of our input image whereby all the pixels relevant for prediction are denoted by 1 and the others as 0. Figure 1 shows LIME masks for FedAVG (ii) and our method (iii) on a test image from the COLORED MNIST dataset (i). Clearly, the former focuses solely on the background to make the model prediction. However, the latter uses causal features in the image (like shape, stroke, edges, and curves) along with some pixels from the background (noise) to make predictions. This demonstrates the reasoning behind robust generalization of FL GAMES as opposed to state-of-the-art FL techniques.

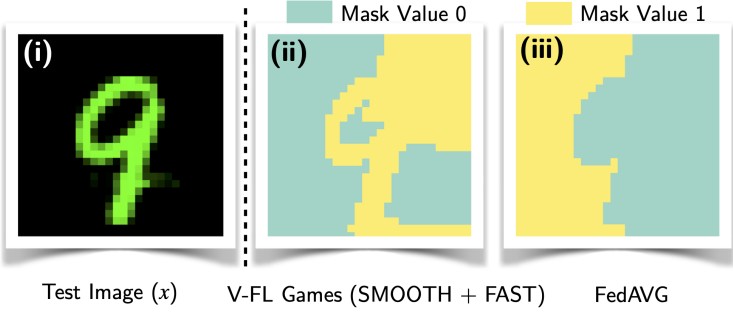

Figure 1: (i) Test image used as input to the interpretable model; (ii) LIME mask corresponding to model prediction using our method V-FL GAMES (SMOOTH+FAST); (iii) LIME mask corresponding to model prediction using FEDAVG;

## 5.3 Ablation Analysis

In this section, we analyze the effect of each of our algorithmic modifications using illustrative figures and computational experiments on the COLORED MNIST dataset. The results on other datasets are similar and are discussed in the supplement.

### 5.3.1 Effect of Simultaneous BRD

We examine the effect of replacing the classic best response dynamics as in Ahuja et al. (2020) with the simultaneous best response dynamics. For the same, we use a more practical environment: (a) more clients are involved, and (b) each client has less data. Similar to Choe et al. (2020), we extended the COLORED MNIST dataset by varying the number of clients between 2 and 10. For each setup, we vary the degree of spurious correlation between 70% and 90%) for training clients and merely 10% in the testing set. A more detailed discussion of the dataset is provided in the supplement. For F-FL GAMES, it can be observed from Figure 2(a), as the number of clients in the FL system increase, there is a sharp increase in the number of communication rounds required to reach equilibrium. However, the same does not hold true for *parallelized* F-FL GAMES. Further, *parallelized* F-FL GAMES is able to reach a comparable or higher test accuracy as compared to F-FL GAMES with significantly lower communication rounds (refer to Figure 2(b)).

### 5.3.2 Effect of a memory ensemble

As shown in Figure 3(left), compared to F-FL GAMES, F-FL GAMES (SMOOTH) reduces the oscillations significantly. In particular, while in the former, performance metrics oscillate at each step, the oscillations in the latter are observed after an interval of roughly 50 rounds. Further, F-FL GAMES (SMOOTH) seems to envelop the performance curves of F-FL GAMES. As a result, apart from reducing the frequency of oscillations, F-FL GAMES (SMOOTH) also achieves higher testing accuracy compared to F-FL GAMES. The

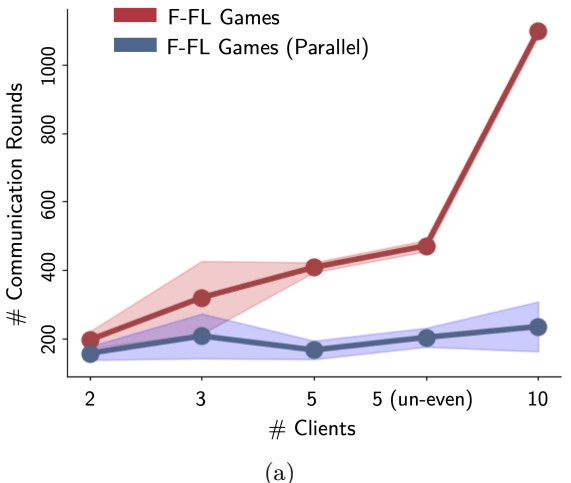

| Type | # clients | Train Accuracy | Test Accuracy |
|------|-----------|----------------|---------------|
| Sequential | 2 | $53.81 \pm 4.14$ | $65.68 \pm 1.89$ |
| | 3 | $54.9 \pm 4.37$ | $66.33 \pm 1.24$ |
| | 5 | $57.20 \pm 2.07$ | $66.53 \pm 0.55$ |
| | 5 (uneven) | $58.09 \pm 2.21$ | $65.30 \pm 2.08$ |
| | 10 | $59.39 \pm 1.41$ | $66.57 \pm 1.02$ |
| Parallel | 2 | $57.95 \pm 3.46$ | $66.57 \pm 2.99$ |
| | 3 | $59.67 \pm 5.46$ | $65.35 \pm 3.73$ |
| | 5 | $61.82 \pm 4.29$ | $65.53 \pm 3.85$ |
| | 5 (uneven) | $56.96 \pm 5.61$ | $66.15 \pm 3.95$ |
| | 10 | $55.24 \pm 2.88$ | $67.49 \pm 3.02$ |

(a)                                            (b)

Figure 2: COLORED MNIST: (a) Number of communication rounds required to achieve Nash equilibrium versus the number of clients in the FL setup; (b) Comparison of F-FL GAMES and F-FL GAMES (Parallel) with increasing clients in terms of training and testing accuracy (mean ± std deviation).

Table 3: COLORED MNIST: Comparison of methods in terms of mean number of rounds required to reach equilibrium.

| | | Algorithm | # Rounds |
|---|---|-----------|----------|
| Fixed | Seq. | F-FL GAMES | 224.0 |
| | | F-FL GAMES (SMOOTH) | 175.0 (1.3×) |
| | Par. | F-FL GAMES | 195.4 (1.1×) |
| | | F-FL GAMES (SMOOTH) | **92.2** (2.4×) |
| Variable | Seq. | V-FL GAMES | 544.4 |
| | | V-FL GAMES (SMOOTH+FAST) | 292.6 (1.9×) |
| | Par. | V-FL GAMES | 499.1 (1.1×) |
| | | V-FL GAMES (SMOOTH+FAST) | **225.5** (2.4×) |

observations are consistent across the *parallelized* variants. Further, as observed from Table 3, F-FL GAMES (SMOOTH) and *parallelized* F-FL GAMES (SMOOTH) significantly reduce the number of communication rounds required to reach equilibrium, further underscoring the efficacy of our proposed methodology.

### 5.3.3 Effect of using Gradient Descent (GD) for $\phi$

Communication costs are the principal constraints in FL setup. Edge devices like mobile phones and sensors are bandwidth constrained and require more power for transmission and reception as compared to remote computation. As observed from Figure 3(right), V-FL GAMES (SMOOTH+FAST) is able to achieve significantly higher testing accuracy in fewer communication rounds as compared to V-FL GAMES. Consistent results are also reported in Table 3, where both sequential and parallel variants of V-FL GAMES (SMOOTH +FAST) result in a significant improvement ($\sim 2\times$) in the number of rounds required.

### 5.3.4 Effect of exact best response

FEDAVG provides the flexibility to train communication efficient and high-quality models by allowing more local computation at each client. This is particularly detrimental in scenarios with poor network connectivity, wherein communicating at every short time span is infeasible.

Inspired by FEDAVG, we study the effect of increasing the amount of local computation at each client. Specifically, in F-FL GAMES, each client updates its predictor based on a step of stochastic gradient descent over its mini-batch. We modify this setup by allowing each client to run a few steps of stochastic gradient

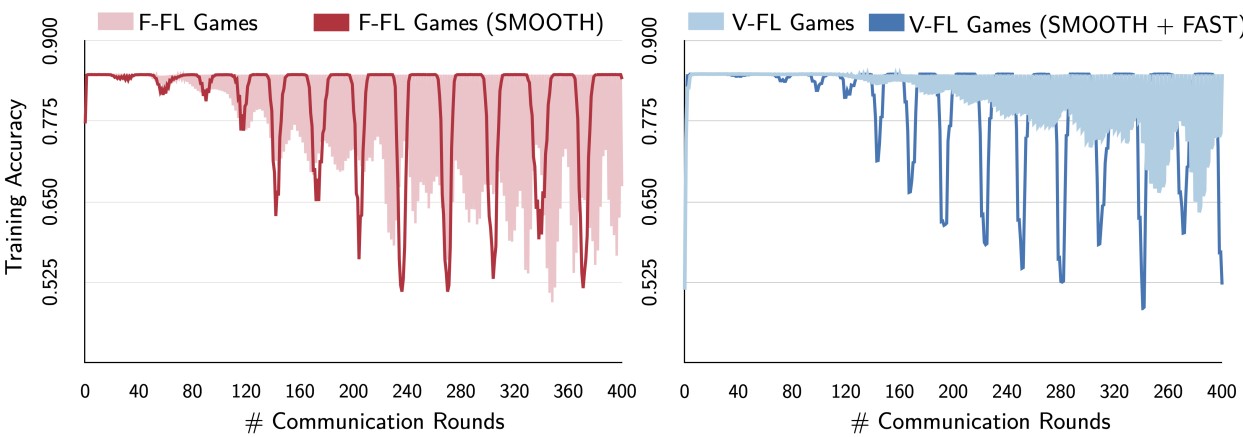

Figure 3: COLORED MNIST: Training accuracy of (left) F-FL GAMES and F-FL GAMES (SMOOTH) for a buffer size of 5; (right) V-FL GAMES and V-FL GAMES (SMOOTH+FAST) with buffer size as 5 versus the number of communication rounds

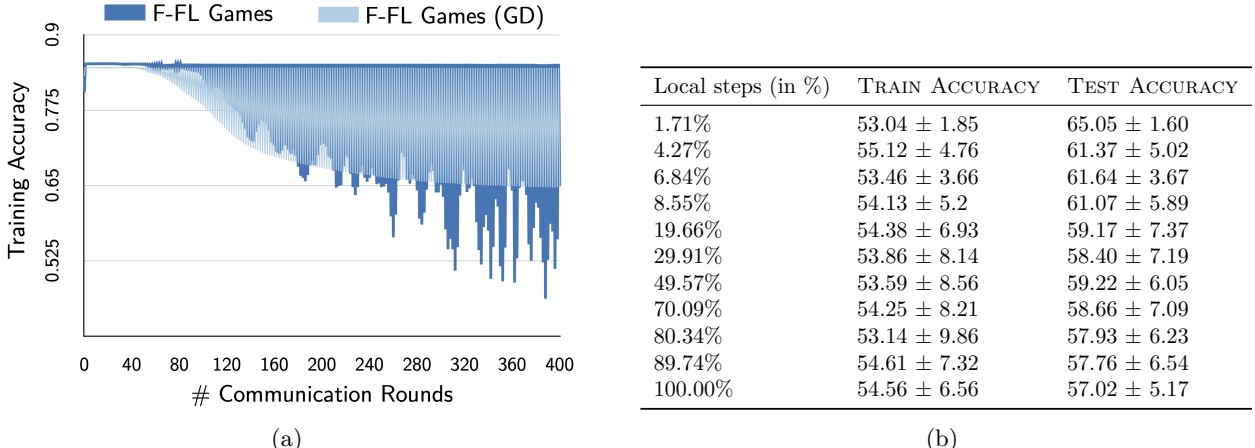

| Local steps (in %) | TRAIN ACCURACY | TEST ACCURACY |
|---|---|---|
| 1.71% | $53.04 \pm 1.85$ | $65.05 \pm 1.60$ |
| 4.27% | $55.12 \pm 4.76$ | $61.37 \pm 5.02$ |
| 6.84% | $53.46 \pm 3.66$ | $61.64 \pm 3.67$ |
| 8.55% | $54.13 \pm 5.2$ | $61.07 \pm 5.89$ |
| 19.66% | $54.38 \pm 6.93$ | $59.17 \pm 7.37$ |
| 29.91% | $53.86 \pm 8.14$ | $58.40 \pm 7.19$ |
| 49.57% | $53.59 \pm 8.56$ | $59.22 \pm 6.05$ |
| 70.09% | $54.25 \pm 8.21$ | $58.66 \pm 7.09$ |
| 80.34% | $53.14 \pm 9.86$ | $57.93 \pm 6.23$ |
| 89.74% | $54.61 \pm 7.32$ | $57.76 \pm 6.54$ |
| 100.00% | $54.56 \pm 6.56$ | $57.02 \pm 5.17$ |

(a)                                      (b)

Figure 4: COLORED MNIST: (a) Effect on Training accuracy of doing gradient descent on each client for updating the predictor versus the standard training paradigm, i.e., F-FL GAMES; (b) Impact of increasing the number of local steps (C) for updating the predictor on the training and testing accuracy (mean $\pm$ std deviation). When the number of local steps (C) reaches 100%, it is equivalent to gradient descent as shown in (a)

descent locally ($C\%$). When the number of local steps at each client reaches is maximum (training data size/ mini-batch size) or $C = 100\%$, the scenario becomes equivalent to gradient descent (GD) over the training data. **[Reviewers' #Y8Gz, #fzQW and #Bwep]:** Since each client's optimal strategy is computed over the entirety of their training data at $C = 100\%$, this scenario corresponds to the exact best response to their opponents. From Table 4(b), it is evident that as the number of local steps increases, the testing accuracy at equilibrium starts to decrease. **[Reviewers' #Y8Gz and #fzQW]:** Since spurious correlations vary considerably across clients in COLORED MNIST, optimal actions for distinct clients differ i.e., one client benefits from positively increasing the correlation between the label and spurious features, while the other benefits from decreasing it. As a result, increasing $C$ in this scenario leads to significant updates that favor one client over the other, making optimization across multiple clients challenging and resulting in reduced performance. When the local computation reaches 100%, i.e. each client updates its local predictor based on a GD over its data, F-FL GAMES exhibits convergence (as shown in Figure 4(a)). FL GAMES is guaranteed to exhibit convergence and good out-of-distribution generalization behavior (Ahuja et al., 2021b) despite increasing local computations. Although the testing accuracy at convergence is lower

compared to the standard setup, this approach opens avenues for practical deployment of the approach in FL.

## 6 Conclusion

In this work, we develop a novel framework based on the Best Response Dynamics (BRD) training paradigm to learn invariant predictors across clients in Federated Learning (FL). Inspired from Ahuja et al. (2020), the proposed method called Federated Learning Games (FL GAMES) learns causal representations which have good out-of-distribution generalization on new train clients or test clients unseen during training. We investigate the high-frequency oscillations observed using BRD and equip our algorithm with a memory of historical actions. This results in smoother performance metrics with significantly lower oscillations. FL GAMES exhibits high communication efficiency as it allows parallel computation, scales well in the number of clients, and results in faster convergence. Future directions include theoretically analyzing the smoothed best response dynamics, which might have potential implications for other game-theoretic machine learning frameworks.

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
