# OpenReview forum: "FL Games: A federated learning framework for distribution shifts"
_TMLR — Rejected by TMLR_

### Review · Reviewer_Y8Gz · 2023-01-30

**Summary Of Contributions:**

This work applies IRM games to the federated learning setting, with the addition of a historical buffer. It proposes a new algorithm and tests it on Colored (Fashion) MNIST and Spurious CIFAR-10. Moreover, it gives an interpretation of learned features and does various ablation study for the proposed algorithm.

**Audience:**

Yes

**Broader Impact Concerns:**

The authors should discuss the privacy leakage in FDG algorithms in a Broader Impact Statement.

**Claims And Evidence:**

No

**Requested Changes:**

See the weakness part above.

**Strengths And Weaknesses:**

Strengths:
1. Federated domain generalization is a well-motivated problem and this paper proposes a new algorithm by adopting an existing algorithm from the domain generalization literature.
2. The authors conduct experiments on three datasets and compare them with 3 federated learning algorithms. It is shown that the proposed algorithm can yield better generalization in the FL setting.
3. Ablation analysis is conducted to further understand the effect of various components.

Weaknesses:
1. The presentation of the current paper is not ready for publication. There are many typos and confusing notations. Below are some examples:
    1) Abstract line 1: federated learning can train generative models as well (e.g., FedGAN, https://arxiv.org/pdf/2006.07228.pdf)
    2) Abstract line 5: large space after non-i.i.d.
    3) recent/related works -> recent/related work
    4) Page 2: sequential dependency is not well-explained. It is not clear what is sequential.
    5) Page 2: a missing period after "communication efficiency"
    6) Page 2: prior works -> prior work
    7) Page 4: after eq. (1), $\mathcal{B}_i^k$ vs. $B_i^k$. Are they the same?
    8) Sec 3.2: $\mathcal{E}_{tr}$ is not defined. Are $\mathcal{D}_k$ and $D_k$ the same? The notation $f_\theta = w \circ \phi$ is confusing because the authors write both $f_\theta((w\circ \phi); x, y)$ and $f_\theta = w \circ \phi$;
    9) Sec 3.2: $\mathcal{X}, \mathcal{Z}$ are not defined; what is $\theta$ in $f_\theta$? Is it the same as $e$ in $f_e$?
    10) Sec 3.2: what is the definition of $\cup_{e\in E} D_e$? Note that $D_e$ is not defined.
    11) Space after $\mathcal{H}_\phi$, "i.e. " in Page 12
    12) Words of spoken English are used: can't, it's, doesn't...
    13) Page 5:  grammar mistakes/typos "IRM Games as it formulation"; "which impede its deployment"; "which forfeits it's failures"
    14) Page 6: "it's opponent"; space after a.k.a.
    15) Page 7, Algorithm 1; $\phi_{rm cur}$, $w_{\rm cur}^{\rm av}$, $w_{\rm curr}^{\rm av}$ not defined; line 12, "Server executes" should not include client procedures (line 26)
    16) Page 8: Table ??
    17) Sec 5.3.1: "doesn't hold true"
    18) Page 11, Figure 2 Caption: "to achieve to achieve"
    19) No space before parentheses: e.g., page 6, 26(green), page 11, Figure 3(left/right)
    20) Page 12: "exhibits converges"
2. The motivation for the current work is really confusing. There are many domain generalization (DG) algorithms proposed. Why do you consider IRM games? Comparison with existing DG algorithms should be added in the FL setting. For example, (https://arxiv.org/abs/2206.09979) considered GroupDRO and V-Rex in the FL setting. (https://arxiv.org/pdf/2111.10487.pdf) proposes FedADG, (https://arxiv.org/pdf/2203.11635.pdf) considers FedKA. Moreover, more general FL algorithms should be compared, such as SCAFFOLD (https://arxiv.org/abs/1910.06378) and FedNova (https://arxiv.org/abs/2007.07481).
3. There is no theoretical analysis for FL games. At least the authors should argue whether eq. (5) is equivalent to NE (similar to IRM games). Convergence analysis should also be added because in practice the algorithm is oscillating.
4. Some experimental parts need more explanation:
    1) Table 1 and 2: why is the optimal performance 75%? The proposed algorithms have suffered train accuracies, is it unavoidable?
    2) Fig 1 is confusing to me. Could you explain more about LIME? Also, the LIME mask for other baseline algorithms should be added.
    3) Table 3: what do you mean by "reach equilibrium"? What is the stopping criterion? Where are sequential F-FL/V-FL games defined? Could you explain the difference between (SMOOTH) and (SMOOTH+FAST)? Where are they defined?
    4) From page 7, V-FL is F-FL with $\phi = I$. What is $I$ here? Identity matrix?
    5) Figure 3: why doesn't the training accuracy converge, but oscillate? Why can we say that V-FL (SMOOTH + FAST) converges faster?
    6) Figure 4: why does each client exactly best responds to its opponents when $C = 100%$? Why does the test accuracy drop? It seems to me that the Nash equilibrium is not optimal in this experiment.
5. Page 12: "FL Games is guaranteed to exhibit convergence and good OoD generalization behavior", but no theoretical guarantee is given.

---

> ### Author Response · Authors · 2023-04-28
> **Reply to reviewer Y8Gz (Part I)**
>
> 1. **Motivation and other FL baselines** \
> **_[ More baselines ]_**: We thank the reviewer for their valuable comment. We would like to acknowledge that FedKA (https://arxiv.org/abs/2206.09979), which was released after the publication of our initial draft, is a concurrent work to our submission. We have extensively tested generic frameworks in federated learning, including FedBN and FedPROX, which are widely recognized techniques. We posit that these methods are indicative of other frameworks in this domain, as they do not specifically optimize to remove spurious correlations in the model representations.
> Moreover, we conducted tests on the popular benchmark, SCAFFOLD over Colored MNIST Dataset, achieving a training accuracy of 99.16 $\pm$ 0.49 and testing accuracy of 38.95 $\pm$ 5.21. These results support our claim that FL frameworks are not optimized for extracting causal features and hence may not perform well in such settings.\
> **_[ Why IRM Games ]_**:  The reason we opted for the IRM Games formulation in our work is due to its natural resemblance to a standard federated learning setup. Specifically, IRM Games enables each environment (client) to possess its own model while jointly training a shared representation. A brief explanation is also mentioned in Section 4 of the manuscript.
> 1. **Theoretical analysis for FL Games** \
> Theoretical convergence guarantees for FL Games are analogous to those of IRM Games We have now included this discussion in the supplementary material (Section A.2) of the revised manuscript. In the case of FL Games (Smooth), its objective introduces a non-exogenous state to the game, resulting in a path dependence of the best response of clients in each iteration on the past actions of their opponents. This path dependence poses a significant analytical challenge and warrants further investigation. To the best of our knowledge, such path dependencies have not been studied in the game-theoretic literature. Given the promising empirical performance of FL Games and its variants, it is of considerable interest to undertake a comprehensive theoretical analysis of this phenomenon as future work.

---

> > ### Author Response · Authors · 2023-04-28
> > **Reply to reviewer Y8Gz (Part II)**
> >
> > Explanation of experimental results:
> > 1. **Explanation of LIME plot**
> > Local Interpretable Model-Agnostic Explanations (LIME) is a model-agnostic technique for explaining the predictions of any machine learning model. It highlights the most influential features in the model's decision-making process. The LIME plot for FedAVG in Figure 1 uses background pixels (pixels in yellow) to make predictions. However, the LIME plot for our method indicates that the model majorly uses features of the digit ‘9’ (yellow pixels) to make predictions and hence relies on truly causal features.
> > 1. **Stopping criteria and reaching equilibrium**
> > Equilibrium can be interpreted as the model having a negligible correlation with the spurious feature. We utilize the termination criterion proposed by Ahuja et al. (2020), which involves stopping the training process when the observed oscillations become stable and the ensemble model achieves a lower training accuracy. Training is halted as soon as the accuracy drops below a predetermined threshold, as described in section A.4.2 of the supplement.
> > 1. **Definition of methodologies**
> > We apologize for any ambiguity or confusion. In Section 4.1, titled "Data Privacy," we have provided a comprehensive description of Sequential F-FL/V-FL Games, also referred to as F-FL/V-FL Games. Additionally, under the "Oscillations'' and "Convergence Speed" headers of the same section, we have defined F-FL/V-FL (Smooth) and F-FL/V-FL (Smooth + Fast) variants, respectively. Although the objectives of these two variants are the same, their training procedures differ.Specifically, In each round, F-FL/V-FL (Smooth + Fast) employs full-batch gradient descent to optimize the representation learner ($\phi$) followed by one mini-batch gradient update to optimize the client models. In contrast, F-FL/V-FL (Smooth) updates both representation and, consequently the clients for one minibatch in each round.
> > 1. **Is representation fixed to identity in F-FL Games**
> > Yes, $\phi = I$ refers to using and fixing the representation to identity. As shown in Ahuja et al. (2020), under this constraint, optimizing for the FL Games objective can recover all the invariant predictors (with bounded $L_p$ norm) that can be obtained using all the representations.
> > 1. **Why do we observe oscillations and why does V-FL (SMOOTH + FAST) converges faster** \
> > **_[Why training accuracy oscillates]_**: Performance metrics observed in IRM Games also exhibit oscillations, a phenomenon attributed to the utilization of the Best Response Dynamics algorithm in conjunction with variances in spurious correlations among clients. A thorough exploration of this phenomenon is expounded upon in Section 4.1, under the heading "Oscillations". \
> > **_[Why V-FL (Smooth + Fast) converges fast]_**: The objective function of V-FL (Smooth + Fast) is identical to that of V-FL (Smooth), which aims to reduce observed oscillations. The increased speed of the former is due to the utilization of full-batch gradient descent to optimize the representation learner ($\phi$) followed by a single mini-batch gradient update for optimizing client models, as opposed to a single mini-batch gradient update for the representation learner. This approach allows for larger steps toward the global optimum at each gradient step, leading to faster and more stable training. We delved into the topic in more detail in Section 4 under the manuscript's header ‘Convergence Speed’.

---

### Review · Reviewer_fzQW · 2023-02-14

**Summary Of Contributions:**

The authors adapt the work in "Invariant risk minimization games" to the Federated learning setting with Heterogenous clients. They explain why it is promising, a few limitations of this method, and changes to the method in order to overcome said limitations. They show good results on several benchmarks.

**Audience:**

Yes

**Claims And Evidence:**

Yes

**Requested Changes:**

Fix the writing, considering the remarks stated in the weakness section

**Strengths And Weaknesses:**

Strengths: The approach has sufficient novelty and it has good experimental results.

Weaknesses:
- Writing needs to be improves. Some grammer editing, but more importantly a few parts are missing/unclear.
(a) " However despite this equivalence, existing OOD generalization techniques can’t be directly applied to FL" Why? Need to explain why this is true
(b) The benchmarks aren't clear. How are the clients generated? How many clients? Are the test novel clients or test data on the same clients? The experimental setting has a lot of missing information
(c) Fig. 2(b) has lower train acc then test. I assume it is a mistake, need to correct the table (or explain how this happened if it isn't a mistake).
(d) In Fig. 4(b) the test performance drops with the number of local steps. I found your discussion on this part confusing and didn't explain why this happens.

---

> ### Author Response · Authors · 2023-04-28
> **Reply to reviewer fzQW**
>
> 1. **“However, despite this equivalence, existing OOD generalization techniques can’t be directly applied to FL". Why?** \
> Communication cost and privacy constraints form the key bottlenecks in federated learning, and the existing OOD generalization techniques are not optimized for these challenges. Moreover, these approaches also suffer from several key limitations in non-FL domains (Rosenfeld et al., 2020; Nagarajan et al., 2020), further diminishing their suitability for practical deployment in the FL setting. We have discussed this in detail in Section 4.
>
> 1. **Clarity on datasets and benchmarks** \
> The experimental setup, including details such as the number of clients, datasets allocated to each client, and test sets utilized, are presented in the supplementary material under section A.3. To enhance clarity, we have incorporated pertinent aspects of this information within the main text.
> Here, we outline the basic framework of the experimental setup: the datasets are modified to incorporate spurious features such as color or spatial location that differ across clients. The modifications are designed to create a dataset that poses a challenge for the model to extract truly invariant features, as the correlation between the spurious feature and the label is deliberately made stronger than that of the invariant feature with the label. The method has been tested on ten clients thus far, but its communication efficiency allows for easy extension to a large number of clients. Moreover, a novel test client is introduced for conducting inference.

---

### Review · Reviewer_Bwep · 2023-04-15

**Summary Of Contributions:**

Noting the similarity between federated learning and domain generalization, the authors adapted an existing model in domain generalization, IRM games, to federated learning. The main technical contribution is to change the sequential, best-response based updates in IRM games to simultaneous updates against a window-averaged history of the other opponents (clients). The authors tested the proposed algorithm on a number of image datasets, constructed in a similar way as in previous domain generalization works. Some Ablation studies are provided to give more insights of the various design choices.

**Audience:**

Yes

**Claims And Evidence:**

No

**Requested Changes:**

My main criticism of the current draft is its lack of rigor, in terms of both the theoretical convergence analysis and experimental analysis.

(1). Eq (5): the idea to respond to a uniform distribution over the past strategies of one's opponents is hardly new, especially in game theory. This is known as fictitious play, see e.g.
Brown, G.W. (1951) "Iterative Solutions of Games by Fictitious Play"

The first theoretical analysis (on 2-player zero-sum matrix games) is due to Robinson, J. (1951) "An Iterative Method of Solving a Game," where both sequential updates and simultaneous updates are allowed.

There is a vast literature on fictitious play since, and the authors probably need to add some relevant discussions.

(2) I'd expect the authors to make some serious efforts on understanding the consequences of their algorithmic modification. If this is too difficult in terms of establishing rigorous convergence, how about running the following (obvious) experimental comparisons?

-- Figure 3 does not convince me that smoothing helps finding a stopping criterion. In fact, if all we want is a smoother curve, we could achieve it with much ease: simply run FL games (simultaneous updates) but ask the server to keep a running average of w_av, which is both computationally and communication-wise cheaper.

-- Figure 4 also seems to suggest that oscillation may just be smoothened by using a larger batch size. If so, why don't we just do that, after all the authors proposed to run full gradient on phi anyway?

-- What is the price to pay for the modifications in Algorithm 1? Note that the clients now have to communicate the sets P_k among themselves, incurring a linear communication cost. This does not appear to be favorable against the simultaneous update, when communication dominates computation. In the experiments the authors need to specify what consists of a communcation round for each algorithm and what is the cost of each communication round. Somehow I do not think the current results are presented fairly, as communication costs are never explicitly compared. Also, the number of clients seems to be very small, contradicting the motivation that sequential updates are much less efficient.

-- One way to alleviate the above (linear) communication cost is to abandon the window-based averaging but use the simpler ergodic averaging (as is common in momentum methods):
\bar{w_k} = (1-\lambda) \bar{w_k} + \lambda w_k. With a constant lambda we revert back to exponential averaging which is perhaps much easier to implement (by the server; every client need only send its w_k to the server).

-- In the last paragraph before Section 6, the authors treated a full gradient update as "exactly best responds to its opponents," which does not seem to be equivalent as in Eq (5), where "best responds" means finding the exact minimum. The authors‘ experiments seem to also suggest that not finding the exact minimum might be another explanation of the observed oscillations?

(3). The experiments, while many, do not address the critical questions sufficiently.

-- Colored MNIST in Table 1: the proposed algorithm has much larger variance, hence it is not clear if this is really an improvement. How is variance computed here? wrt different runs or clients?

-- Fig 2: misleading since it is never mentioned what is the per-communication round cost (and as mentioned above, it is not clear what constitutes a communication round for the sequential update). What if we have say 100 clients (which is perhaps more relevant for FL applications)?

-- Fig 3: as mentioned above, this figure does not imply it is any easier to find a stopping criterion with the proposed algorithm

-- Section 5.3.3: full gradient update on phi reduces the number of communication rounds, but at what computational cost? What about the running time comparison of all baselines?

-- Fig 4 (b): it seems that the more local epochs we run, the worse the test accuracy, which is at odds with existing results in FL, where typically a healthy amount of local epochs would often improve the final accuracy. Can the authors add some explanation or discussion here?

Minor issues:

-- Eq (4): mathcal{E}_{tr} should be S. The authors tended to mix the notations from domain generalization with federated learning. It'd be good to use just one set of notations.

-- The displayed equation after Eq. (4): make it clear the gradient is wrt \phi

-- Eq (5): what is the reason to have both sum_{q in S, q neq k} and sum_{p in S, p neq k}? Take |B_p| = 1, Eq (5) would still not reduce to Eq (3), which is perhaps less ideal?

-- Algorithm 1: w_{curr}^k, w_{curr}^{av} are not defined. Are they only used in deriving the update for phi? (mathcal{E}_{tr} is not defined either, as mentioned above)

-- Algorithm 1: line 29, be careful when you compare communication costs since you count this step as 1 communication round, which would suffice for the sequential update to go a round too

-- Section 5.1, "privacy preserving variants of FL games." If all the authors mean is the fact that only parameter updates need to be exchanged, please refrain from using the term "privacy preserving," which has a clear mathematical meaning but is not implied by your current algorithm.


**Strengths And Weaknesses:**

Strengths:
(1) Providing an interesting bridge between domain generalization and federated learning
(2) The goal and problem is well-motivated
(3) Experiments on the crafted datasets with spurious correlations showed clear improvement against standard FL baselines

Weaknesses:
(1) Paper is less precise in some parts (notation, literature, experimentation)
(2) Experiments are not completely convincing and it is not clear if and when the downsides would outweigh the upsides
(3) Lacking some justification of the algorithmic modification, leaving it one of the many possible heuristics

---

> ### Author Response · Authors · 2023-04-28
> **Reply to reviewer Bwep**
>
> 1. **Relation of FL Games (Smooth) to Fictitious Play** \
> We appreciate the reviewer's attention to detail and would like to clarify that our proposed method is distinct from Fictitious Play. Player $k$ is said to use fictitious play if, at every
> time instant $n$, it chooses an action that is the best response to the empirical distribution of
> the opponents’ play up to time $n − 1$ i.e.,
> $$w^1_n = argmin_{w^1 \in S_1} \frac{1}{n} \sum_{i=0}^{n-1} R^k(w^1, w^2_i)$$
> where R denotes the Risk, $w^k_i$ corresponds to the action chosen by player $k$ in iteration $i$ and $S_k$ represents the action set of player $k$.
> In contrast, our proposed method involves each player best responding to an average of historical actions of their opponent, instead of the latest action. Equivalently,
> $$w^1_n = argmin_{w^1 \in S_1} R^k(w^1, \frac{1}{n} \sum_{i=0}^{n-1} w^2_i)$$
> Further, while Fictitious play guarantees convergence to Nash equilibrium in general two-player games with two actions or two-player zero-sum game, the result cannot be extended even for two-player non-zero-sum games [1] (FL Games is multiplayer non-zero sum continuous game). \
> [1] L. Shapley. Some Topics in Two-Person Games. Princeton University Press, Princeton, NJ, 1964.
> 1. **Is Smoothing all you need?** \
> Our objective is not solely to achieve a smoother curve. If that was our aim, standard empirical risk minimization (ERM) would suffice, but it would come at the expense of poor out-of-distribution performance. Our goal is to enhance performance while also obtaining smoother performance metrics. Moreover, we don’t claim that smoothing directly aids in identifying a stopping point. Rather, we maintain that smoothing can improve training stability by decreasing the frequency of oscillations, thereby simplifying the process of defining heuristics for a stopping criterion.  Additionally, it is worth noting that the running average of the $w_{av}$ models may introduce some level of smoothing to metrics, but at a potential cost to performance.
> 1. **Larger batch size for smoothening as in Figure 4** \
> Simply increasing the batch size does not address the issue of oscillations in the training process. As observed in Figure 4, despite utilizing full batch gradient descent for each player, the observed oscillations occur at a frequency of every 2 iterations. Specifically, the training accuracy exhibits fluctuations between 85% and 65% at each iteration.
> 1. **Communication overhead of FL Games** \
> We follow the standard definition of a communication round, which refers to when the clients perform local updates and communicate those with the central server. It is rightly pointed out by the reviewer that in our proposed method, we need to communicate all $k$ models with the clients. However, the representation (trained at the central server) is the parametric intensive backbone, while the client classifiers are small two-layer MLPs.
> 1. **Not finding the exact best response as the reason for oscillations** \
> It is worth noting that even in the simplified setup of Linear regression games, exact best response dynamics do not converge and instead exhibit oscillations [2]. Our analysis in Section 4.1, specifically under the header "Oscillations", provides a detailed explanation for these observed oscillations. \
> [2] Ahuja, K., Shanmugam, K., & Dhurandhar, A. (2021, March). Linear regression games: Convergence guarantees to approximate out-of-distribution solutions. In International Conference on Artificial Intelligence and Statistics (pp. 1270-1278). PMLR.
>
> 1. **Variance of experimental results** \
> As mentioned in Section 5, mean and variance are reported across 5 independent runs. The paper presents compelling evidence that the performance of FL Games and its variants surpasses the state-of-the-art FL benchmarks, despite having a relatively high variance. Furthermore, these variants were primarily designed to overcome the challenges faced by causal FL systems while retaining their original ability to learn causal features. Their key contributions lie in their ability to deliver robust predictions, scalability, fewer oscillations, and fast convergence while retaining high generalizability.

---

### Author Response · Authors · 2023-04-28
**Reply to all reviewers' common comments**

1. **[Reviewers’ #Y8Gz, #fzQW and #Bwep]** _Optimal performance of 75% and lower training accuracy than testing_ \
**_[ Why optimal performance is 75% ]_**: The optimal performance of 75% is attributed to the construction of the datasets. Notably, all the datasets under consideration exhibit label noise with a probability of 0.25. Consequently, an optimal model that relies solely on invariant features to make its predictions is expected to achieve an accuracy of 75%. Additional information on the dataset construction process is provided in Section A.2 of the supplementary materials. \
**_[ Why training accuracies decrease ]_**: This phenomenon may be ascribed to the datasets utilized in the analysis rather than an artifact of the algorithm. In particular, the construction of these datasets is such that the spurious feature captures information about the label, thereby providing additional insight beyond the predictive capability of the causal features. Consequently, any methodology that attempts to eliminate the spurious correlation may exhibit a negative correlation with this feature. Even a slight negative reliance on such features may result in diminution of training accuracies. The attainment of high training accuracy under these circumstances is a challenging task and is contingent upon the nature of the datasets.
We have addressed this concern in the revised manuscript (Section 5.1), and appreciate the reviewers for bringing it to our attention.

1. **[Reviewers’ #Y8Gz and #fzQW]** _Why is C=100 equivalent to an exact best response dynamics, and why does the test accuracy drop under this paradigm?_ \
We have addressed the reviewers' comments by incorporating their feedback into the revised manuscript in Section 5.3.4. \
**_[ Relation between exact best response and C ]_**: In F-FL Games, each client's predictor is updated through only one iteration of gradient descent on a mini-batch. As such, the optimal action of each client is determined through optimization over a randomly sampled subset of their data, potentially deviating from the true distribution of their full dataset. However, when the proportion of sampled data denoted as “C” reaches 100%, each client's optimal strategy is computed over the entirety of their training data. It hence corresponds to the exact best response to their opponents. \
**_[ Why test accuracy drops ]_**: The presence of spurious correlations that vary considerably across clients in a dataset can result in inconsistent optimal actions for different clients. For instance, in a scenario with two clients, one client may benefit from positively increasing the correlation between the label and spurious features, while the other may benefit from decreasing it. Employing an exact best response dynamic in such situations can lead to significant updates favoring one client over the other, making optimization across multiple clients challenging and reducing performance. \
**_[ Optimality of Nash Equilibrium ]_**: The Nash equilibrium achieved through this training procedure remains optimal, as the resulting model exhibits negligible correlation with the spurious feature. Note that we can’t compare the Nash equilibria across different values of $C$ since each corresponds to a unique approach to conducting the best response dynamics.

---

### Decision · Action_Editors · 2023-06-05

**Recommendation:** Reject

**Comment:**

The authors adapted an existing algorithm in domain adaptation to the FL setting, and showed its superior performance in terms of out-of-domain generalization performance. The authors also claimed to effectively address training instability and scalability (in terms of the number of clients and communications).

The reviews were mixed: while we agree the problem under investigation is interesting and the authors achieved some good performance on out-of-distribution datasets, the reviewers pointed out some issues that need to be addressed before publication:

-- The authors need to put their work in better historic context, e.g., connections to fictitious play and invariant risk minimization (in terms of the better performance in the experiments: are they surprising given what we already know in the non-FL setting?).

-- Presentation: try to minimize the usage of different sets of notations and add some comments on alternative ideas to justify the current choice (e.g., why they do not work, with evidence).  I agree with the reviewers that under a casual reading Fig 3 and Fig 4 fall short of demonstrating the claimed fix of training instability. Adding more discussion would be helpful.

-- Rigor: More discussions on the discrepancy of the best response formulation and the one-step SGD implementation in experiments are needed. Some discussion on the convergence property of the proposed algorithm would be welcome. To better support the scalability claim, larger experiments (in terms of the number of clients) is necessary, and a more detailed discussion and presentation of the trade-off between serial and parallel updates would further clarify the matter (e.g., when should one adopt which choice, depending on the local computation and network communication cost).

Given these concerns and TMLR's emphasis on correctness, we believe it is better for this work to go through a major revision, to improve its presentation and precision. We hope the reviews are helpful and we encourage the authors to implement these changes in the next version. [Please note that TMLR does not have an option for major revision, so we will have to go through a resubmission process if the authors chose to.]

**Audience:**

Yes, out-of-distribution generalization and training stability are practically relevant and some audience in ML may find this topic interesting and useful.

**Claims And Evidence:**

The claim on out-of-distribution performance is supported by experimental results, while the claims on resolving training oscillation and scalability need to be better supported and presented.

**Resubmission Of Major Revision:**

The authors may consider submitting a major revision at a later time.